# High-performance particulate matter including nanoscale particle removal by a self-powered air filter

Guo-Hao Zhang[1], Qiu-Hong Zhu[1], Lei Zhang[1], Fang Yong[1], Zhang Zhang[1], Shuang-Long Wang[1], You Wang[1], Ling He[1✉] & Guo-Hong Tao [1✉]

Particulate matter (PM) pollutants, including nanoscale particles (NPs), have been considered serious threats to public health. In this work, a self-powered air filter that can be used in high-efficiency removal of PM, including NPs, is presented. An ionic liquid–polymer (ILP) composite is irregularly distributed onto a sponge network to form an ILP@MF filter. Enabled by its unique electrochemical properties, the ILP@MF filter can remove $PM_{2.5}$ and $PM_{10}$ with high efficiencies of 99.59% and 99.75%, respectively, after applying a low voltage. More importantly, the charged ILP@MF filter realizes a superior removal for NPs with an efficiency of 93.77%. A micro-button lithium cell or silicon-based solar panel is employed as a power supply platform to fabricate a portable and self-powered face mask, which exhibits excellent efficacy in particulate removal compared to commercial masks. This work shows a great promise for high-performance purification devices and facile mask production to remove particulate pollutants.

[1] College of Chemistry, Sichuan University, 610064 Chengdu, China. ✉email: lhe@scu.edu.cn; taogh@scu.edu.cn

A long with economic development and industrialization advancement, the demand for fossil fuel energy, such as coal or petroleum is growing, resulting in the rapid increase in particulate matter (PM) pollution[1]. According to their aerodynamic diameters, PM can be grouped as $PM_{10}$ (with a diameter ranging from 2.5 to 10 μm), $PM_{2.5}$ (with a diameter ranging from 1 to 2.5 μm), and nanoscale particles (NPs, with a diameter <1 μm). The major chemical constituents in PM, such as transition metal oxides, inorganic salts and organic carbons, have been acknowledged to have a severe impact on human health[2–4]. NPs, with smaller diameters and larger surface areas, are suspended for longer in the air and result in more harm to the public[5,6]. A large number of studies have found that widespread exposure to NPs is nocuous to the main organs of humans, such as the respiratory tract, skin, brain, lungs, and immune system[7–10], leading to neuroinflammation and decreased cognitive abilities, even causing a DNA damage[11–15]. According to an analysis of the Global Burden of Disease Study 2017 (ref. [16]), ambient particle pollution, including nanoparticle pollution, was one of the top four risk factors contributing to death and disability-adjusted life years at the national level in China, with the second highest change (88.5%) from 1990 to 2017. Therefore, the development of removal technology for NPs is of considerable significance.

The capture of NPs by conventional purification technologies, which focus on a size-dependent mechanism, is quite difficult due to their ultra-low mass and small particle size[17]. Traditional filters with regular pores act as an obstacle to the particle-transport path, and the filters can effectively intercept larger particles. However, for NPs with diameters smaller than the pore size, some particles may follow the gas streamline and pass through the purification material, resulting in a less effective filtration performance[18,19]. To resolve this limitation, a series of functional materials and production technologies based on top-down fibre manufacturing processes have been developed[20–23], of which Brownian diffusion was surmised to be a dominant mechanism[24–26]. For instance, the Wang group reported creating a rotating triboelectric nanogenerator-enhanced polyimide nanofibre air filter by electrospinning[21]. The nanogenerator charged the fibre film to create a high voltage electric field and therefore enhanced the removal efficiency of NPs. They achieved improved performance for NP removal without increasing the pressure drop. However, the method requires independent units of a nanogenerator and fibre membrane, which is inconvenient for fabricating portable devices. Lee's group applied an Al-coated conductive fibrous filter to capture NPs by strong electrostatic forces in the fibres[23], which realized a high efficiency of over 99.99%. However, the specialized instrument and ultra-high voltage may consume large amounts of energy and ionize air, thus producing ozone and leading to further health risks. Therefore, constructing acceptable high-performance NP removal materials without requiring high voltages remain a challenge.

Ionic liquids (ILs), consisting entirely of ions, are extensively studied in areas of electrochemistry and electrochemical devices due to their properties of a wide electrochemical window, high electric conductivity and electrochemical stability[27–30]. Recently, IL–polymer (ILP) composites have shown great performance in the field of electrochemistry, such as transparent solid electrodes, self-healing ionic conductors, conductive biofibres, etc.[31–35]. Generally, upon the addition of ILs into polymers, there will be two effects: first, the ILs will plasticize the polymers and endow the ILP composites with specific mechanical properties[31]; second and more importantly, the ILs induce a significant conductivity enhancement so that the ILP composites can generate a high electric field under a low voltage[32], which results in the attraction of particulates, especially NPs, which are smaller and lighter than

$PM_{2.5}$ and $PM_{10}$. In addition, a previous study has reported that ILs possess high absolute electrostatic potential (ESP) values[36], which influence the motion of particles in air and allow a capability for particle removal through electrostatic interactions. The ILP composites should therefore be promising for use in high-efficiency and low-energy-consumption air cleaning devices without employing high voltages. However, to the best of our knowledge, there is no research on such composites as conductive materials for PM removal.

Herein, we report a self-powered air filter based on ILP composites with excellent hydrophilicity, high ESP and admirable conductivity for PM and NP removal. The ILP composites are prepared by metal/halogen-free IL 1-alkyl-3-methylimidazolium acetate ($[C_n mim][OAc]$) and hydrophilic polymers, including poly(acrylamide) (PAM), poly(vinyl alcohol) (PVA) and poly(vinylpyrrolidone) (PVP). To create a desirable partial porous structure, the ILP composites are irregularly coated onto the skeleton of a melamine-formaldehyde (MF) resin sponge to construct an ILP@MF filter. This specific structure is capable of allowing polluted air to flow adequately in the sponge channels, resulting in a reduced pressure drop and enhanced removal efficiency. The ILP@MF filter system can be used to collect PM by generating a high electric field with a low voltage of 3 V, which can be supplied by a small button lithium cell or silicon-based solar panel. Therefore, excellent filtration functionality is obtained with the face mask manufactured using the ILP@MF filter without an external power supply, indicating that the ILP@MF filter has great potential to produce self-powered and wearable cleaning devices.

## Results

A schematic illustration of the fabrication process for the ILP@MF multilayer porous air filter is shown in Fig. 1a. A solution of $[C_n mim][OAc]$ was added to the ethanol solution of the pre-dispersion polymer under stirring conditions. Then, the MF sponges were immersed in the prepared ILP solution. Vacuum pumping was employed to remove the entrapped ethanol and gas in the sponge. More details of the methods employed are included in the "Methods" section near the end of the paper. Figure 1b illustrates photographs and scanning electron microscopy (SEM) images of the bare MF sponge and ILP@MF filter. Compared with the initial sponge, the ILP composite films are generated and irregularly distributed on the multi-layered scaffold of the MF sponge. Interestingly, only a portion of the holes are coated by the ILP composite on each single layer, as seen in the schematic diagram and the magnified SEM images of the ILP@MF filter. As the illustration shows in Fig. 1c, this specific structure can allow the polluted air to flow adequately in the sponge channels, and the particles are fully in contact with the ILP composite, which may lead to low air resistance and efficient capture.

Fourier transform infrared (FTIR) spectra and powder X-ray diffraction (PXRD) patterns (Fig. 2a–d) were used to characterize the structure of the pure polymers, $[C_4 mim][OAc]$ and ILP composites with different amounts of IL ($m_{IL}/m_{polymer}$ = 0.2, 0.5 and 1). In the FTIR spectra, the strong absorption peak at 1564 $cm^{-1}$ belongs to the stretching vibration of the C = N group of the $[C_4 mim]$ cation, which enhances as the content of $[C_4 mim]$ [OAc] increases. The characteristic peaks of PAM, PVA and PVP are observed in the spectra of the ILP composites at 1061, 1723 and 1648 $cm^{-1}$, respectively. FTIR spectra of different ILs and polymers are examined (Supplementary Fig. 1), and similar peaks were obtained for the ILP composites. PXRD data reveal that the diffraction peaks of $[C_4 mim][OAc]$–PAM are sharp and intense, indicating its highly crystalline nature (Fig. 2d). The ILPs

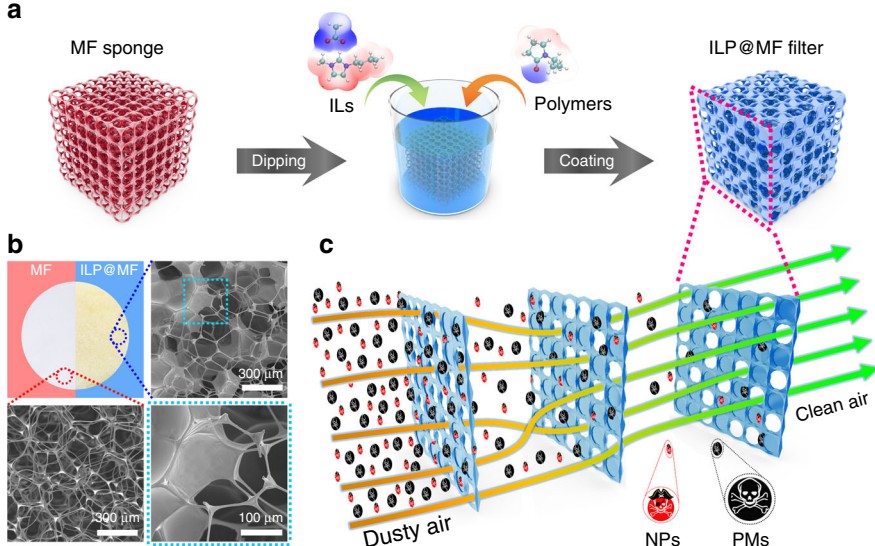

**Fig. 1 Fabrication method and purification model of ILP@MF filter. a** Schematic representation of the dipping-coating progress for the fabrication of the ILP@MF filter. **b** Photographs and SEM images of the MF sponge (red) and the ILP@MF filter (blue). **c** Illustration of the removal mechanism of the multilayer ILP@MF filter.

consisting of PVA and PVP are amorphous (Supplementary Fig. 2). Thermogravimetric analysis (TGA) and differential thermal analysis (DTA) results illustrate that all the ILP composites are thermodynamically stable, with decomposition temperatures higher than 205 °C (Fig. 2e, f).

Generally, PM in air contains a large amount of atmospheric moisture, and the moisture acts as an adhesive to all metal ions and fine organic carbon, forming a complex system[3,37]. Therefore, good hydrophilicity is beneficial for the affinity of composites towards PM complexes[38]. The hydrophilicity of the ILP composites with different amounts of [C$_4$mim][OAc] was estimated based on the water contact angle, and the results are presented in Fig. 2g. The water contact angle of the pure PVP film is ca. 81°, while a water contact angle of ca. 66° is obtained for [C$_4$mim][OAc]–PVP (0.2). With increasing amounts of inserted [C$_4$mim][OAc], the water contact angle values exhibit a downward trend, decreasing from ca. 50° ([C$_4$mim][OAc]–PVP (0.5)) to ca. 41° ([C$_4$mim][OAc]–PVP (1)). When more [C$_4$mim][OAc] was added, we found that the ILP composites were not case hardened, resulting in the inability to form a filter. Moreover, similar decreases in the water contact angle are observed in the ILP composites of PAM and PVA. The water droplet has a time-dependent decrease and finally infiltrates the film (Fig. 2h). The surface free energy for the ILP composites above was calculated by the Owens two-zone mechanism of adhesion (Supplementary Table 1), and the results coincide with the contact angle experiments. The conclusion above indicates that the ILP composites are sufficiently hydrophilic materials, which are predicted to remove particles in air.

Due to the outstanding hydrophilicity that the ILP composites exhibit, the PM removal performance of different ILP@MF filters was tested. Figure 3a shows a schematic diagram of the designed testing device that investigates the PM filtration performance. The flow rate, pressure drop ($\Delta P$) and concentration of PM were determined by commercial detectors. The removal efficiency ($\eta$) was calculated using the following equation:

$$\eta = \left(1 - \frac{C_{out}}{C_{in}}\right) \times 100\% \qquad (1)$$

where $C_{in}$ and $C_{out}$ are the mass concentrations ($\mu g\, m^{-3}$) of the particles before and after removal, respectively[39,40].

We first examined the removal efficiencies of ILP@MF filters with ILP composites consisting of PVP and different ILs (Fig. 3b). The average PM$_{2.5}$ removal efficiencies for the ILP@MF filters consisting of [C$_4$mim][OAc], [C$_6$mim][OAc] and [C$_8$mim][OAc] are 98.78 ± 0.42%, 97.53 ± 0.51% and 96.21 ± 0.53%, respectively. For PM$_{10}$, the removal efficiencies are 99.02 ± 0.48% for [C$_4$mim][OAc], 98.64 ± 0.41% for [C$_6$mim][OAc] and 97.47 ± 0.44% for [C$_8$mim][OAc]. It is clear that the shorter the length of the alkyl group is, the better the performance in the removal test. We further investigated the filtration performance among the ILP@MF filters with three kinds of polymers. Compared with ILs, the variety of polymers has a greater impact on the removal ability of ILP@MF filters, and the removal efficiencies of the PAM, PVA and PVP composites are 92.14 ± 0.63%, 86.27 ± 0.61% and 98.81 ± 0.57% for PM$_{2.5}$ and 94.98 ± 0.59%, 88.31 ± 0.57% and 99.09 ± 0.49% for PM$_{10}$, respectively. The results demonstrate that the PVP-based filter possesses the best removal ability of the tested polymers, which can be attributed to the differences in the morphology and hydrophilicity of the filters (Fig. 2g and Supplementary Fig. 3). Thus, the ratio of components of [C$_4$mim][OAc]–PVP (0.2, 0.5 and 1) was discussed in the PM removal tests. Among them, [C$_4$mim][OAc]–PVP (1) shows the highest removal efficiency of 98.77 ± 0.46% for PM$_{2.5}$ and 99.07 ± 0.54% for PM$_{10}$.

To further study the removal mechanism of the composite, we explored the surface ESP of the PVP and [C$_4$mim][OAc]–PVP composite by the density functional theory (DFT) method, where a high absolute ESP value indicates a great influence on the motion of PM[36,41–43]. Figure 3c depicts the ESP-mapped van der Waals (vdW) surface of the simplified structure of PVP and [C$_4$mim][OAc]–PVP. Compared with pure PVP, [C$_4$mim][OAc]–PVP exhibits a higher global maximum and minima for ESP on the surface at +48.72 and −62.95 kcal mol$^{-1}$, which are electrostatic enhancements of 166.08% for the maximum and 37.24% for the minima, respectively. In addition, [C$_4$mim][OAc]–PVP shows a vast vdW surface area (73.6 A$^2$) featuring high absolute ESP values (<−30 and >+30 kcal mol$^{-1}$), which are much larger than those of pure PVP (13.7 A$^2$). The above results imply that the ILP@MF filter of [C$_4$mim][OAc]–PVP has the greatest potential for air filtration applications.

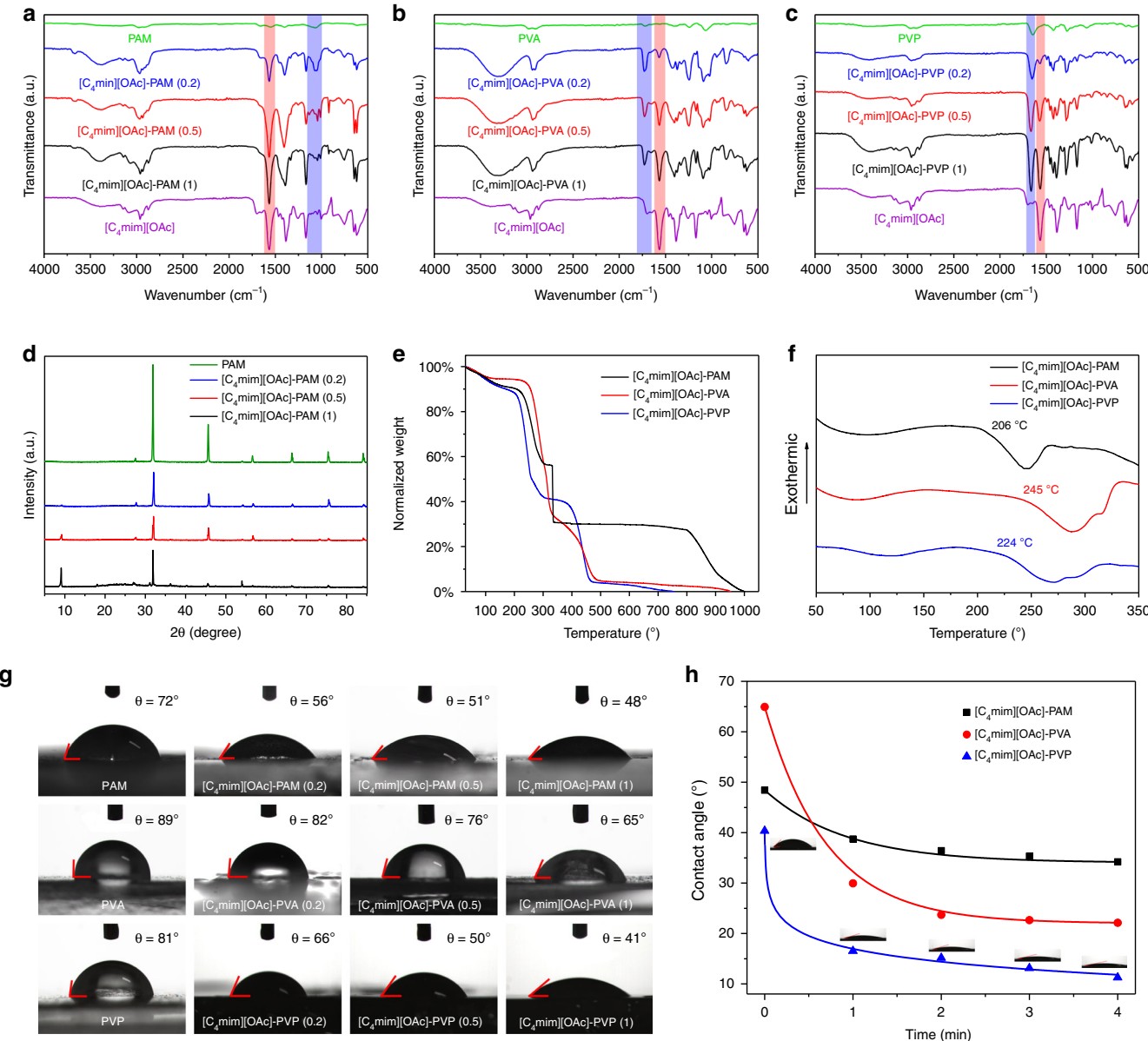

**Fig. 2 Basic properties characterization of ILP composites.** FTIR specra of the ILP composites consisting of [C$_4$mim][OAc] and **a** PAM, **b** PVA, **c** PVP with different amounts of ILs. **d** PXRD patterns of the ILP composites consisting of [C$_4$mim][OAc] and PAM. **e** TGA and **f** DTA curves of the ILP composites with different polymers. **g** Water contact angle of the ILP composites consisting of [C$_4$mim][OAc] and different polymers with different amounts of ILs. **h** Time dependence of the contact angles of water droplets on the different ILP films. Source data are provided as a Source Data file.

The admirable properties and performance of the [C$_4$mim][OAc]–PVP@MF filter inspired us to investigate its capture capacity after applying a voltage. The electrochemical stability of the [C$_4$mim][OAc]–PVP composite was first researched by cyclic voltammetry (CV) (Supplementary Fig. 5). A wide electrochemical window higher than 3.1 V is observed, indicating that the [C$_4$mim][OAc]–PVP composite is a stable material under an applied voltage of 3 V. Electrochemical impedance spectroscopy (EIS) was used to measure the charge transport parameters of the composite, such as the sheet resistance ($R_S$) and charge conductivity ($\sigma$). The Nyquist plots of the pure [C$_4$mim][OAc] and [C$_4$mim][OAc]–PVP composite are presented (Supplementary Fig. 6). [C$_4$mim][OAc], whose initial $R_S$ value is 5.8 $\Omega$ sq$^{-1}$, offers good conductivity to the composites with an $R_S$ of 13.9 $\Omega$ sq$^{-1}$. Furthermore, we define the conductive ability of the [C$_4$mim][OAc]–PVP composite by calculating the $\sigma$ value according to the

equation below:

$$R_S = \frac{L}{A\sigma} \qquad (2)$$

where $A$ presents the active film area and $L$ presents the thickness[44]. Pure PVP has poor conductivity with a $\sigma$ value[45] lower than $10^{-8}$ mS m$^{-1}$. Compared with PVP, a conductivity value of $1.03 \times 10^2$ mS m$^{-1}$ was obtained for the [C$_4$mim][OAc]–PVP composite. Therefore, the [C$_4$mim][OAc]–PVP composite is a desirable conductive material that can produce a high electric field to capture PM, especially NPs.

Figure 4a shows a sketch of the filtration test with an applied voltage. The variable applied voltage and electric field distribution are provided by a numerical control transformer. Two pieces of copper electrodes were inserted at the ends of the [C$_4$mim][OAc]–PVP@MF filter, so that the [C$_4$mim][OAc]–PVP

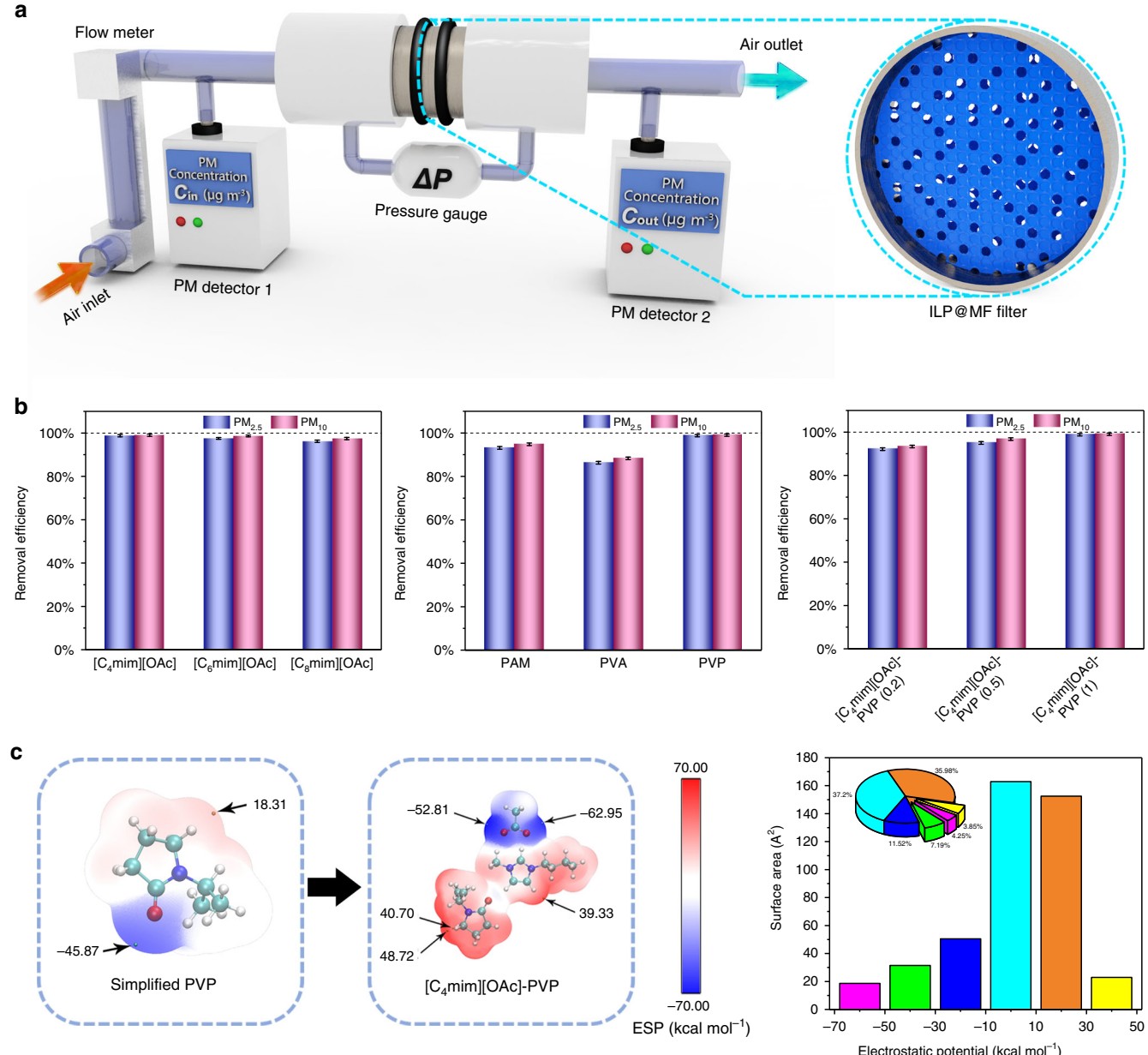

**Fig. 3 Particulate matter purification test and performance comparison among different ILP@MF filters. a** Schematic diagram of the device for the removal experiment and the measurement of air flow rate, pressure drop, and PM concentrations. **b** Removal efficiencies of the ILP@MF filters with different ILs, polymers, and amounts of [C₄mim][OAc]. Error bar represents the standard deviation of three replicate measurements. **c** ESP-mapped molecular vdW surface of the simplified PVP, [C₄mim][OAc]–PVP and the surface area in each range of the [C₄mim][OAc]–PVP. Source data are provided as a Source Data file.

composites distributed on the sponge can contact the electrode surface to form a closed loop. Figure 4b shows the fractional removal efficiencies under various applied voltages from 0 to 3 V. Compared to the uncharged [C₄mim][OAc]–PVP@MF filter, the removal efficiencies of PM₂.₅ increased as the applied voltage increased to 3 V (99.59 ± 0.31%). Similar phenomena were also obtained for PM₁₀, whose removal efficiency increased from 99.01 ± 0.23% (0 V) to 99.75 ± 0.22% (3 V). Notably, an obvious improvement in the capture for NPs is observed, as the green points show. When the imposed voltage is 0 V, the removal efficiency of the uncharged [C₄mim][OAc]–PVP@MF filter for NPs is 35.12 ± 0.69%, which can be attributed to the great hydrophilicity and ESP values of the [C₄mim][OAc]–PVP

composite. With the increase in the applied voltage, the removal efficiencies exhibit a significant enhancement. The removal efficiency of NPs can be up to 93.77 ± 1.05% when the imposed voltage rises to 3 V. Furthermore, the removal results of PM were confirmed by SEM and dynamic light scattering (DLS). SEM images show that the [C₄mim][OAc]–PVP@MF filter changes significantly compared to the original, with agglomerated PM₂.₅ and PM₁₀ particles deposited on the surface of the filter (Fig. 4c, red and blue frames). With a closer look, the nano-sized particles with nonuniform diameters ranging from dozens to hundreds of nanometres are adhered on the surface of the filter film (Fig. 4c, green frame). The DLS curves further prove the removal efficiency of the proposed filter for NPs in a wide size range

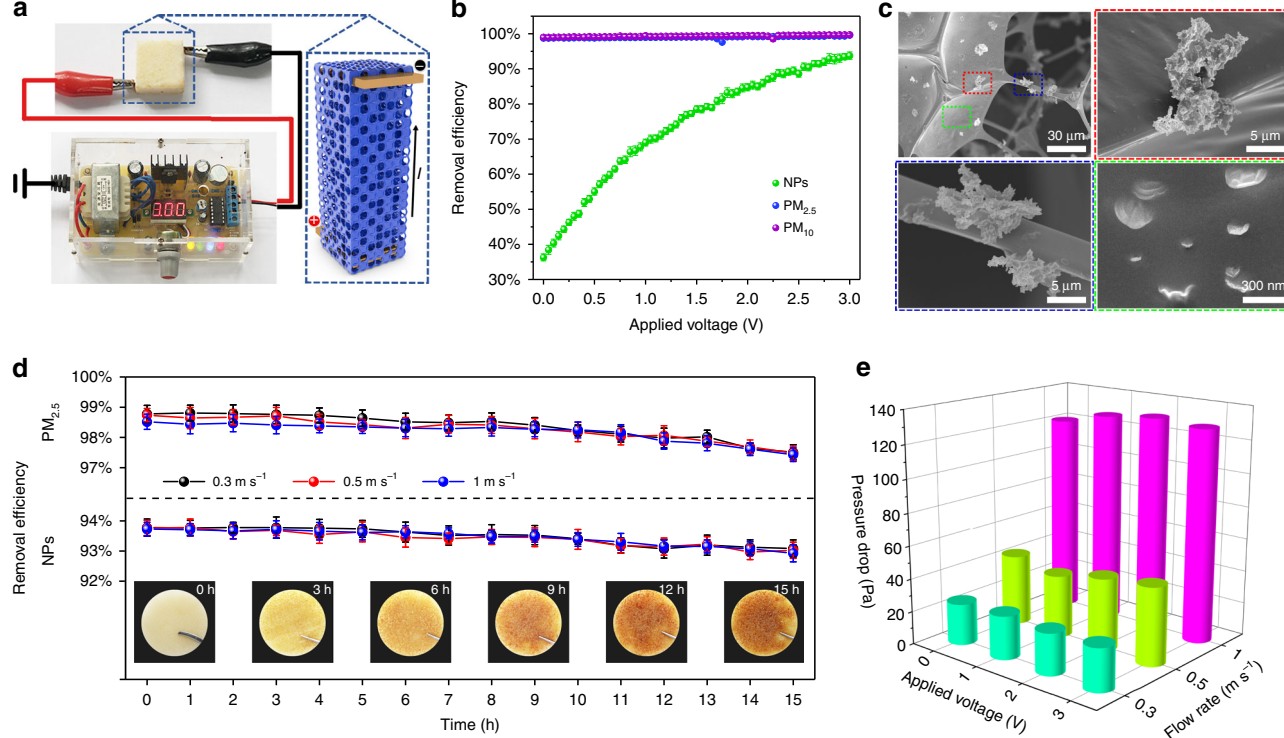

**Fig. 4 Performance enhancement of ILP@MF filter after applying voltage. a** Schematic of the charged [C$_4$mim][OAc]–PVP@MF filter. **b** Efficiency points of [C$_4$mim][OAc]–PVP@MF filter for NPs, PM$_{2.5}$ and PM$_{10}$ under different voltage. **c** SEMs of the [C$_4$mim][OAc]–PVP@MF filter after a 15 h removal experiment. Error bar represents the standard deviation of three replicate measurements. **d** 15 h test of the removal efficiency for PM$_{2.5}$ and NPs of the [C$_4$mim][OAc]–PVP@MF filter. The inset images are photos of the [C$_4$mim][OAc]–PVP@MF filter during the purification test. **e** Pressure drop of the [C$_4$mim][OAc]–PVP@MF filter under various applied voltage (0, 1, 2, and 3 V) and at different air flow rate (0.3, 0.5, and 1 m s$^{-1}$). Source data are provided as a Source Data file.

(Supplementary Fig. 7). The composition of the filter after the removal test was probed by energy dispersive spectroscopy (EDS) spectra and mapping images (Supplementary Figs. 8 and 9). Compared to the original filter (C, O and N), some other metallic elements, such as Ca, Fe, K and non-metallic elements, such as Si, P, and S, are found, which means that particles derived from burned cigarettes are effectively captured. The observations above demonstrate that applying a voltage to the filter is beneficial for removing PM, particularly NPs.

Then, we measured the removal performance of the [C$_4$mim][OAc]–PVP@MF filter for 15 h under an applied voltage of 3 V. As shown in Fig. 4d, the removal efficiencies for PM$_{2.5}$ and NPs remained at high levels of over 97% and 92%, respectively, after the filtration test. Moreover, the removal efficiencies of the [C$_4$mim][OAc]–PVP@MF filter had no significant change as the air flow rate increases from 0.3 to 1 m s$^{-1}$. The photographs of the filters during the purification test are shown in the inset of Fig. 4d. After 15 h of PM removal, the colour of the [C$_4$mim][OAc]–PVP@MF filter changed from white to brown, indicating uptake of the particles in polluted air. The pressure drop ($\Delta P$) is one of most important parameters in the performance of a filter. A low $\Delta P$ value indicates a desirable air flow resistance. The pressure drop between the inlet and outlet of the [C$_4$mim][OAc]–PVP@MF filter was measured under various applied voltages and flow rate conditions (Fig. 4e). The average pressure drop of the filter with a 3 V voltage is only 26 Pa at a flow rate of 0.3 m s$^{-1}$, which is far lower than the fine standard of the U.S. Department of Energy (ca. 325 Pa at an air flow rate of 5 cm s$^{-1}$)[46].

The filter consisting of ILP composites and MF sponges can be simply regenerated. The regeneration procedure for the ILP@MF filter is illustrated (Supplementary Fig. 10). The images after

regeneration and the removal efficiencies of the regenerative [C$_4$mim][OAc]–PVP@MF filter are shown in Fig. 5a. A conspicuous colour variation is noticed in the inserted photographs before and after regeneration, owing to cleaning of the surface covered by dusty particle pollutants. The SEM images further demonstrate that almost all the particles are successfully removed by the regeneration process. Furthermore, the regenerated [C$_4$mim][OAc]–PVP@MF filter still maintains a high level capture ability with removal efficiencies for NPs of over 90% and over 97% for PM after being recycled 10 times, indicating that the [C$_4$mim][OAc]–PVP@MF filter possesses a reliable regeneration capacity.

A wearable and self-powered face mask consisting of a [C$_4$mim][OAc]–PVP@MF filter was designed and fabricated, as depicted in Fig. 5b. Benefiting from the excellent removal properties of the [C$_4$mim][OAc]–PVP@MF filter after applying a voltage, a 3 V button lithium cell was installed on the face mask as a portable power supply platform to obtain a self-powered system. The employed voltage of 3 V, which is below the secure voltage (12 V) required by the International Electrotechnical Commission[47], is considered to be absolutely harmless to the human body. The quality factor (QF) is calculated according to the equation below:

$$QF = \frac{-\ln(1 - \eta)}{\Delta P} \quad (3)$$

where $\eta$ represents the removal efficiency and $\Delta P$ represents the pressure drop[48,49]. Generally, a higher QF value indicates better behaviour of a filtration device (i.e., higher removal efficiency and lower pressure drop)[36,40]. The filtration performance of the face masks based on the [C$_4$mim][OAc]–PVP@MF filter and the

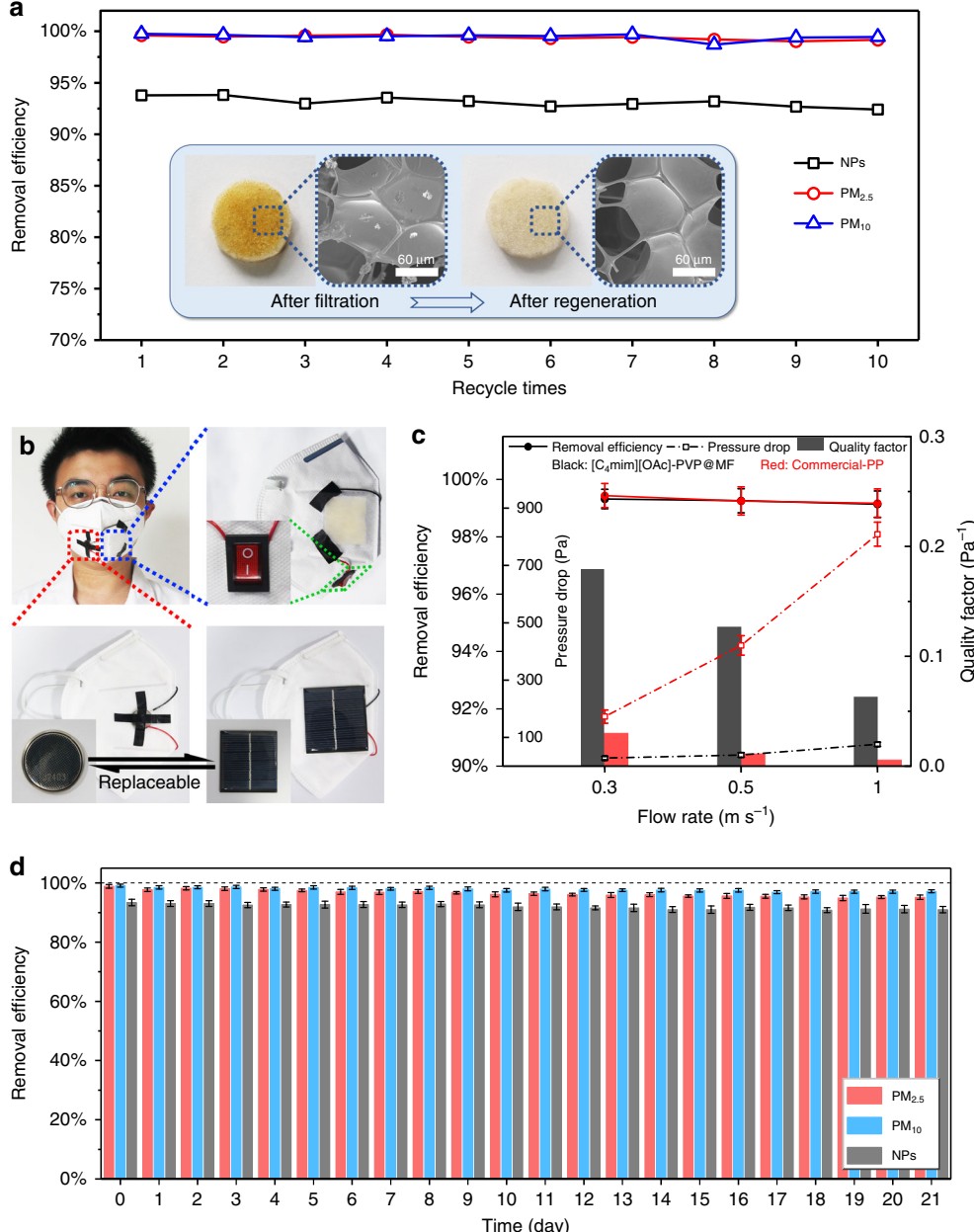

**Fig. 5 Renewability test and filtration performance comparison of ILP@MF masks and commercial face masks. a** Removal efficiency of the charged [C$_4$mim][OAc]–PVP@MF filter regerated for 1–10 times. The inset images are photo of the filter before and after regeration. **b** The photographs of wearable and self-powered face mask made from [C$_4$mim][OAc]–PVP@MF filter. The button lithium cell and the silicon-based solar panel are replaceable to each other. **c** Comprehensive index comparison including removal efficiency, pressure drop, and quality factor between masks made of [C$_4$mim][OAc]–PVP@MF filter (black lines and pillars) and commercial-PP filter (red lines and pillars) under different flow rate. Error bar represents the standard deviation of three replicate measurements. **d** Long-term PM removal performance. Source data are provided as a Source Data file.

commercial PP filter were compared under high concentration particulates (above 300 μg m$^{-3}$)[39], whose diameters ranged from dozens of nanometres to tens of micrometres. Figure 5c shows a comparison of the comprehensive index between the [C$_4$mim][OAc]–PVP@MF filter and commercial PP filter. Both the [C$_4$mim][OAc]–PVP@MF filter and the commercial filter exhibit excellent removal performance for PM with the highest efficiencies being over 99%. However, the commercial PP filter suffers from a drastically increasing pressure drop with an increase in the air flow rate, leading to low QF values (0.030 Pa$^{-1}$ at 0.3 m s$^{-1}$ and 0.006 Pa$^{-1}$ at 1 m s$^{-1}$). The mask based on [C$_4$mim][OAc]–PVP@MF emerges as desirable with average

pressure drops (28 Pa at 0.3 m s$^{-1}$ and 76 Pa at 1 m s$^{-1}$), which results in admirable QF values (0.178 Pa$^{-1}$ at 0.3 m s$^{-1}$ and 0.063 Pa$^{-1}$ at 1 m s$^{-1}$). Interestingly, the button lithium cell that supplied the voltage source for the mask can be replaced by a silicon-based solar panel. The mask consisting of solar panel performs well as the mask made with the button cell (see Supplementary Fig. 11 for details), which means that the face mask can directly harness solar energy to realize PM purification without other power sources. Moreover, a switch is devised and inserted at the bottom of the mask to provide users the ability to employ the mask according to the actual atmosphere. When people are in a mild pollution environment, the switch-off mode

can be chosen; when people are in a heavily polluted environment, the switch-on mode can be selected. Thus, the service life of the mask can be prolonged.

Stability is of great significance for practical applications of materials. Therefore, we further performed a continuous filtration test for over 3 weeks to explore the long-term stability of the mask manufactured with ILP@MF filters. As shown in Fig. 5d, the removal efficiencies for $PM_{2.5}$ and $PM_{10}$ are $98.86 \pm 0.61\%$ and $99.10 \pm 0.51\%$ (0 day), $96.98 \pm 0.73\%$ and $98.03 \pm 0.44\%$ (7th day), $95.94 \pm 0.63\%$ and $97.53 \pm 0.56\%$ (14th day), $95.24 \pm 0.49\%$ and $97.04 \pm 0.55\%$ (21st day), respectively. The results show that the ILP@MF filter still maintained high removal efficiencies for $PM_{2.5}$ and $PM_{10}$ after 21 days. In addition, the purification ability for NPs was also retained (>90%). The measurement results show that the ILP@MF filters possess considerable stability for long-term PM purification.

## Discussion

In this work, we proposed a wearable and self-powered air filter based on an ILP composite and MF resin porous sponges. Due to the enhanced hydrophilicity and favourable electrochemical properties, the obtained $[C_4mim][OAc]$–PVP@MF filter exhibits high removal efficiencies of 99.59% for $PM_{2.5}$ and 99.75% for $PM_{10}$, respectively, after applying a low voltage of 3 V. Even more importantly, the $[C_4mim][OAc]$–PVP@MF filter realizes an enhanced removal capacity for NPs with a prominent efficiency of 93.77%. Moreover, the specific coating structure of the filter leads to a low pressure drop in the whole filtration system.

Profiting from the desirable filtration performance, wearable and self-powered face masks based on ILP@MF were designed and fabricated. Table 1 exhibits the filtration performance and costs comparison between the ILP@MF face masks and the commercial masks. Compared to the commercial filters, the fabrication costs of the ILP@MF masks are higher, but the ILP@MF masks exhibit superior filtration performance for NPs, including a low pressure drop (<27 Pa) and high QF (>0.100 $Pa^{-1}$) at a flow rate of 0.3 m s$^{-1}$, while the values for the commercial filter are higher than 173 Pa and below 0.004 $Pa^{-1}$, respectively. A button lithium cell or silicon-based solar panel with a switch can be employed as a portable power supply platform. By controlling the switch, the mask can work in different modes under different environmental conditions. Therefore, the ILP@MF filter is promising for manufacturing wearable and self-powered air-cleaning devices with excellent removal performance.

## Methods

**Preparation of ILP composites.** Acetate ILs 1-alkyl-3-methylimidazolium acetate ($[C_nmim][OAc]$) were synthesized through a two-step method according to the literature procedures[50,51]. In the first step, 1-methyl imidazolium and alkyl chloride were stirred at 70–75 °C for 100 h to synthesise 1-alkyl-3-methyl imidazolium chloride ($[C_nmim]Cl$). In the second step, $C_nmim]Cl$ and $NH_4[OAc]$ were mixed in acetone and stirred at room temperature for 48 h to form $[C_nmim][OAc]$. The NMR spectra of $[C_nmim][OAc]$ are showed (Supplementary Figs. 13–15). $[C_4mim][OAc]$: $^1H$ NMR (400 MHz, DMSO-$d_6$, δ) = 0.82 (s, 3H), 1.18 (s, 2H), 1.60 (s, 2H), 1.73 (s, 2H), 3.89 (s, 3H), 4.21 (s, 3H), 7.89 (s, 1H), 7.98 (s, 1H), 10.07 (s, 1H) ppm;

$^{13}C$ NMR (101 MHz, DMSO-$d_6$, δ) = 13.22, 18.80, 25.78, 31.58, 35.49, 122.37, 137.78, 174.08 ppm. $[C_6mim][OAc]$: $^1H$ NMR (400 MHz, DMSO-$d_6$, δ) = 0.82 (s, 3H), 1.22 (s, 6H), 1.59 (s, 3H), 1.76 (s, 2H), 3.88 (s, 3H), 4.19 (s, 2H), 7.83 (s, 1H), 7.91 (s, 1H), 9.98 (s, 1H) ppm; $^{13}C$ NMR (101 MHz, DMSO-$d_6$, δ) = 14.22, 22.35, 25.62, 26.32, 29.93, 31.04, 35.95, 49.00, 122.71, 124.00, 138.08, 174.10 ppm. $[C_8mim][OAc]$: $^1H$ NMR (400 MHz, DMSO-$d_6$, δ) = 0.81 (s, 3H), 1.20 (s, 9H), 1.60 (s, 3H), 1.76 (s, 2H), 4.19 (s, 2H), 7.86 (s, 1H), 7.94 (s, 1H), 10.07 (s, 1H) ppm; $^{13}C$ NMR (101 MHz, DMSO-$d_6$, δ) = 22.09, 25.57, 25.82, 28.43, 28.57, 29.61, 31.22, 35.47, 48.56, 122.30, 123.58, 137.78, 173.81 ppm. In 20 mL vials, 0.2 eq. (1 mmol), 0.5 eq. (2.5 mmol), and 1 eq. (5 mmol) $[C_nmim][OAc]$ were added into 5 mL volume water solution, respectively. The solutions were sonicated for 5 min. Then 1 eq. (5 mmol) hydrophilic polymers (PAM, PVA or PVP) were added to water solution of $[C_nmim][OAc]$ under stirring. The resulting mixture were sonicated for 30 min at 60 °C. Then the solutions were dried in vacuum under ambient conditions for at least 4 h to yield pale yellow ILP composites.

**Preparation of ILP@MF air filters.** Solution of ILP composites were prepared at room temperature in mixed solvent of water and ethanol (concentration: 100 mg mL$^{-1}$, $V_{(water)}:V_{(ethanol)} = 3:2$). The porous scaffold MF sponges with thickness of 4 mm were cleaned successively with ethanol in an ultrasonic bath for 20 min, followed by drying overnight at 60 °C in an electric oven. Then the sponges were dipped into the solution of ILP composites and sonicated for 20 min. After that, the porous composites were dried in vacuum at 65 °C for 2 h and 85 °C for 2 h, respectively.

**Filtration test.** The sample PMs were produced by the smoke burned from cigarette with particles in the size range from <50 nm to >10 μm, which were detected by DLS and laser particle analysis (Supplementary Figs. 16 and 17). The particle size distribution is analogous to available literature data[21,52,53]. The smoke PMs were adjusted by injecting air and controlling the gas flowing rate of 0.3–1 m s$^{-1}$. The $PM_{10}$, $PM_{2.5}$ and NPs concentration were adjusted by injecting air and controlling the gas flowing rate which made the index about 1000 μg m$^{-3}$. PM counts were measured using a BR-HOL-1210 laser PM detector. Pressure drop (ΔP) were measured by differential pressure AS8520 from SMART SENSOR.

**Instrument and characterization.** FTIR were obtained from Bruker ALPHA infrared spectrometer in the range of 400−4000 cm$^{-1}$. PXRD was recorded on Schimadzu-XRD-6100 at a scanning rate of 5° min$^{-1}$. $^1H$ and $^{13}C$ NMR spectra were taken on Bruker 400 MHz nuclear magnetic resonance (NMR) spectrometer operating at 400 and 100 MHz, respectively, with $d_6$-DMSO as the locking solvent. The $^1H$ and $^{13}C$ chemical shifts are reported in ppm relative to TMS. TGA and DTA was performed using a Shimadzu DTG-60H thermal analyzer. Measurements were accomplished by heating the samples at a heating rate of 10 °C min$^{-1}$ from ambient temperature to 1000 °C. DLS were recorded on Zetasizer Nano ZS90 with 12 cycle scans for every time, three times for every test. Laser particle size measurement were recorded on HELOS-RODOS/M. CV and EIS were characterized by a DH7000 electrochemical workstation with a scanning rate of 0.1 V s$^{-1}$. Contact angle images were recorded on a ZHONGCHEN JC2000DS contact angle meter equipped with a CCD camera. SEM and EDS mapping images were obtained by using a JSM-7500F scanning electron microscope.

**Theoretical study.** The structure of N-isopropyl pyrrolidone (simplified structure of PVP), and $[C_nmim][OAc]$ were optimized in gas phase at B3LYP/6-311+ +G** level. The quantitative analysis of molecular surface for the compounds above at the B3LYP/6−311+ +G(d, p) level was performed by the Gaussian09 (Revision A.02) suite of programmes[54]. The Gaussian output wfn files were used as inputs for Multiwfn to perform the quantitative analysis[41]. The colour mapped isosurface graphs of ESP were rendered by VMD 1.9.3 programme[55]. The vdW surface referred throughout this paper denotes the isosurface of $r = 0.001e$ bohr$^{-3}$. The quantitative analysis of molecular surface is significant to study noncovalent

**Table 1 Filtration performance summary of ILP@MF masks compared with commercial face masks.**

| Mask type | PSP | E (%) | ΔP (Pa) | QF (Pa$^{-1}$) | Costs (RMB) |
|---|---|---|---|---|---|
| ILP@MF-1 | Button cell | 93.37 | 26 | 0.104 | 1.5 |
| ILP@MF-2 | Solar panel | 93.21 | 27 | 0.100 | 2.8 |
| Commercial-1 | – | 42.06 | 173 | 0.00315 | 0.8 |
| Commercial-2 | – | 18.77 | 423 | 0.000646 | 0.2 |

PSP power supply platform, E NPs removal efficiencies, ΔP pressure drop, QF quality factor, ILP@MF-1 ILP@MF mask fabricated by button lithium cell, ILP@MF-2 ILP@MF mask fabricated by silicon-based solar panel, Commercial-1 commercial PP face mask, Commercial-2 commercial cotton face mask. Filtration performance was tested at the flow rate of 0.3 m s$^{-1}$.

interaction. ESP can be written as equation below[42]

$$V_{Total}(r) = V_{Nuc}(r) + V_{Eles}(r) = \sum_A \frac{Z_A}{\sqrt[2]{(r-R_A)^2}} - \int \frac{\rho(r')}{\sqrt[2]{(r-r')^2}}dr' \quad (4)$$

where $Z$ and $R$ denote nuclear charge and nuclear position, respectively. Then the $V_S^+$, $V_S^-$ and $V_s$, denoting average of positive, negative and overall ESP on vdW surface, respectively, can be expressed as equations below

$$V_S^+ = \left(\frac{1}{m}\right)\sum_{i=1}^m V(r_i) \quad (5)$$

$$V_S^- = \left(\frac{1}{n}\right)\sum_{j=1}^n V(r_j) \quad (6)$$

$$V_S = \left(\frac{1}{z}\right)\sum_{k=1}^z V(r_k) \quad (7)$$

where $i$, $j$ and $k$ are indices of sampling points in positive, negative and entire regions, respectively, $t$ is the total number of surface vertices. A positive (negative) value means that current position is dominated by nuclear (electronic) charges.

## Data availability
The data that support the findings of this study are available from the corresponding author upon reasonable request. The source data underlying Figs. 2a–f, h, 3b, c, 4b, d, e, 5a, c and d and Supplementary Figs. 1a–c, 2a, b, 4a–d, 5–8, 11, 12a–d, 16 and 17 are provided as a Source Data file.

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

## Acknowledgements

The finance support of National Natural Science Foundation of China (No. 21876120), and China Scholarship Council are gratefully acknowledged. We also thank the Comprehensive training platform of specialized laboratory, College of chemistry, Sichuan University and the Analytical & Testing Center of Sichuan University for instrumental measurement.

## Author contributions

L.H. and G.-H.T. conceived and supervised the research. G.-H.Z. and G.-H.T. proposed the idea and designed the experiments. S.-L.W. and Y.W. tested the composite and collected the data. L.Z. and Z.Z. co-analysed the results. G.-H.Z. and Q.-H.Z. fabricated the face masks. F.Y and Q.-H.Z. performed the filtration test of filters and masks. G.-H.Z., Q.-H.Z. and F.Y. wrote the paper.

## Competing interests

The authors declare no competing interests.
