## [Peer Review File · Nature Communications]

Reviewers' comments:

Reviewer #1 (Remarks to the Author):

Review of High-Performance Particulate Matter and Nanoscale Particle Removal by Self-powered Air Filter

This paper presents the concept and preliminary performance data for a new portable air filter that can be used as a personal face mask. The concept uses a melamine-formaldehyde resin sponge as the filter skeleton, which is then coated with an ionic liquid polymer having high hydrophilicity and conductivity. The latter property allows for generation of high electric fields at low voltages, which helps to attract particulates to the filter. Moreover, the low voltage allows for a portable power supply (i.e. lithium cell or solar panel) to be included with the mask.

The authors demonstrate over 99% removal of particulate matter (typically in the 1.0 – 10 micron size range) and over 93% removal of nanoscale particles (less than 1.0 micron in diameter). Moreover, the filter is easily regenerated, though becomes discolored. After 10 uses, the removal of particulate matter remains above 97% and the removal of nanoscale particles remains above 90%.

Overall, I found this paper to present a novel subject, which may be of broad interest to the readers of Nature Communications. Air pollution is a widespread concern, especially near industrial cities without tight regulations. If the proposed technology is easy and inexpensive to implement for the consumer market, and gives superior performance to the common face masks used today, then it could have broad impact.

A few comments and questions aired at improving the paper are

1. There are many grammatical errors and awkward sentences.

Answer: We appreciate the reviewer for taking the time to review our manuscript and valuable comments. The grammatical errors and awkward sentences have been corrected in the revised manuscript. The revised manuscript was edited and corrected by the editors from Nature Research Editing Service.

2. The paper organization is nonconventional. For example, the Introduction spans lines 22-328, yet only lines 22-78 are introductory or background material. The rest are results of the study, which should be a separate Results section, with many subsections. Further, the Methods are given at the end, rather than before the Results (which may be okay, if pointed out in a short description of the paper organization at the end of the Introduction).

Answer: According to your suggestion. The results of the study containing lines 92-238 have been separate to an independent “Results” section, following the “Introduction” part. Besides, the “Methods” section has been placed at the end of the “Discussion” section.

3. In line 44, it is stated that nanoscale particles are difficult to remove by size-based mechanisms. However, wouldn't Brownian diffusion be a dominant mechanism for such small particles?

Answer: Yes, we agree with the point that Brownian diffusion is a major mechanism for small particles, especially the nanoscale particles. It is because of the motions like Brownian diffusion of the small particulates that they can be captured by filters, as mentioned in the literatures “*Adv. Funct. Mater.* **2018**, 28, 1706680”, “*ACS Nano*, **2015**, 9, 12552”, “*J. Mater. Chem. A*, **2015**, 3, 23946”, and “*ACS Appl. Mater. Interfaces* **2016**, 8, 8086”, etc. We sincerely apologize that we did not state accurately in the preliminary manuscript. Corresponding description has been corrected in the revised manuscript.

4. It would be helpful to discuss how the new filter compares to existing technologies and standards. Are the removal efficiencies similar to or better than those common cotton face masks? How might the costs compare?

Answer: This is a constructive suggestion. In the preliminary manuscript, we have already compared the filtration performance of the ILP@MF filter and the commercial PP filter. According to Chinese standard for air-purifying particle respirator “GB 2626-2006” and American standard announced by National Institute of Occupational Safety and Health (NIOSH), both the filters can be grouped as KN95 (China) or N95 (America) level for PM_{2.5} and PM₁₀ filtration. We further compared the filtration performance and the costs of the face masks fabricated by ILP@MF, PP fiber and cotton. As shown in Supplementary Fig. 12a-c, filtration performance for PM_{2.5}, PM₁₀ and NPs of the three masks at the flow rate of 0.3 m s⁻¹ were tested. Besides, Supplementary Fig. 12d exhibits the pressure drops of the masks at different flow rates. The results show that the ILP@MF filter exhibits high removal efficiencies (>98%) for PM_{2.5} and PM₁₀ as good as PP filter, which is higher than the cotton filter (around 80% for PM_{2.5}, 90% for PM₁₀). As for NPs, the ILP@MF filter presents the highest removal efficiency of over 93%, while the PP and cotton filters are around 40% and 20%, respectively. Furthermore, we compare the manufacturing cost of ILP@MF filter and the two commercial masks. The results were presented in the Table below.

Mask type	Filter costs	Power supply platform costs	Total costs
ILP@MF	0.5 RMB	Button lithium cell: 1 RMB	1.5 RMB
ILP@MF	0.5 RMB	Silicon-based solar panel: 2.3 RMB	2.8 RMB
PP mask	0.8 RMB	∖	0.8 RMB
Cotton mask	0.2 RMB	∖	0.2 RMB

Reviewer #2 (Remarks to the Author):

In this manuscript, the authors proposed a self-powered air filter (ionic liquid-polymer@MF) for PM removal. Though the fabrication approach of ILP@MF filter is well presented, PM removal with ionic liquid modified nanostructure had been studied by previous literature (Jing, et al, *ACS Appl. Mater. Interfaces* **2016**, 8, 7030–7036). Moreover, the referee thinks that there are basic conceptual and technical flaws in this manuscript regarding the characterization of filtration performance. Besides, not enough data has been provided by the authors to support their claim. Thus, in the current form of this paper, I do not suggest the work to be published in Nature Communications. Some important issues need to be addressed first as shown below.

*Answer: We feel great thanks for your review work on our article. We agree with that the ionic liquid modified nanostructure has been studied in the field of particulate purification. Our previous works are also engaged in the relevant research (*J. Mater. Chem. A*, **2019**, 7, 4619-4625; *Adv. Mater. Interfaces* **2018**, 5, 1700448). In this work, we propose a novel concept of self-powered air filter using ionic liquid to form high electric fields at low voltages and corresponding preliminary purification data, which has never been studied in the previous works. The proposed technology is easily applicable, and exhibits a better filtration performance comparing to the existing masks in market.*

1. The authors cited the Ref [1] to support “the demand of fossil energy like coal or petroleum is growing, resulting in the rapid increase of particulate matter (PM) pollution”. However, in that referenced paper, it only said that fossil fuel consumption can result in the greenhouse gas emissions. No information about PM can be found in that reference paper.

Answer: Before manuscript submission, the “Introduction” section has been modified many times. The previous version of this sentence is “The booming of industry requires numerous usages of coal or petroleum, which are to blame for the sharply deteriorating atmospheric environment”. What we firstly wanted to express was that the use of fossil fuels would have an impact on the environment, so we cited the Ref [1]. In the further revision of the manuscript, we changed the sentence but forgot to change the corresponding reference. We apologize for this mistake. The reference of the sentence was corrected in the revised manuscript. Besides, the remaining references have been further verified.

2. In the title, it is written that “Particulate Matter and Nanoscale Particle Removal”. Actually, nanoparticles belong to particulate matter. Also, it is written that “Particulate matter (PM) pollutant including nanoscale particle (NP) have been considered as serious threats to public health”. Since the authors know that NPs are part of PM, how can they still make PM and NP in parallel in many parts of the manuscript? This kind of conceptual mixing is very confusing. The referee thinks that the authors also made a mistake on the definition of PM. Volatile organic compounds are another type of the

pollutants, not the components of PM. The authors made several such kind of concept mistakes in the manuscript, which shall be corrected.

Answer: As the reviewer have pointed out, the nanoscale particle (NP) does belong to particulate matter (PM) group. In the preliminary manuscript, the intention of listing NPs separately from PMs was to emphasize the importance of NPs purification and the significance of our work. But the statements may be misleading by readers. We have corrected in the revised manuscript.

For your second comment. At first, during the researching of literatures, we found that there are literatures considering the airborne PM as a highly complex and heterogeneous mixture of chemical and/or biological elements including volatile organic compounds or organic (carbonaceous) aerosols “*Toxicology in Vitro* **2009**, 23, 37-46; *Atmos. Environ.* **2011**, 45, 5655-5663”. The VOC was seemed as a part of PM in the preliminary manuscript because that it is considered as an important kind of precursor of PM. Some secondary organics with lower vapor pressure in VOC can form secondary organic aerosol (SOA) through nucleation, condensation, gas particle distribution and other processes, and become part of PMs (*J. Geophys. Res. Atmos* **2003**, 108, 51; *Atmos. Environ.* **2013**, 71, 260-294). Herein, our aim was to highlight the harmfulness of PM. For the avoidance of doubt, we have modified the corresponding statements in the revised manuscript.

3. The filtration test in the manuscript looks problematic. The filtration efficiency could be very high with this testing method, which may have low accuracy. The authors only tested the PM concentration after filtration. According to schematic diagram of the device for the particle removal experiment, PM particles is likely to leak out before entering the pipe. Therefore, the PM concentration before removal seems to be smaller than $1000 \mu\text{g m}^{-3}$. The authors should detect the real-time PM concentration before and after filtration at the same time. The authors are suggested to refer previous literature for the basic testing process, such as. Liu et al, *Nat. Comm.* **2015**, 6, 6205; Han et al, *Adv. Func. Mater.* **2019**, 1903633; Wang et al, *ACS Appl. Mater. Interfaces* **2016**, 8, 23985–23994.

Answer: As you have pointed and the previous literatures mentioned, the real-time concentration detection for air inlet and outlet is convictive and necessary, so as we did in this work. We already gave out the determination detail in the previous manuscript (line 158 to 162), the statement of the equation for calculating removal efficiencies (η) notes that C_{in} and C_{out} are the mass concentrations ($\mu\text{g m}^{-3}$) of particles before and after removal. Without the C_{in} and C_{out} data, we could not obtain any removal efficiency result in the manuscript. The schematic diagram of the filtration device was drawn by a professional drawing company, who designed the device picture to achieve a better aesthetic feeling. To show the testing process in more details, a more detailed diagram of filtration device has been replaced in the revised manuscript.

4. For PM size larger than 1 micron, mass concentration of PM detector is accurate. For nanoparticles, however, mass concentration can hardly be precisely obtained due

to their very small mass. Thus, the measurement error would be high since the same PM detector was applied to detect the nanoparticles and fine/coarse particles.

Answer: Indeed, we agree with that better instruments can get more accurate data for NPs detection. We sincerely feel sorry that the detecting equipment used in our lab is limited. But it is noted that our experimental results have a good repeatability. Each efficiency data was obtained through three parallel determinations and the standard deviation was within a controllable range. Besides, by comparison experiments with other commercial filter, the ILP@MF filter does have an enhanced performance on NPs purification. Therefore, we think the data has a certain reference value for promising the concept of the self-powered air filter.

5. Different polymers can result in quite different filtration efficiency, why the variety of polymers have such great impact on this performance?

Answer: The species of polymers have a high impact on filtration performance of ILP@MF filters. To explore this, SEM images of the filters with same ionic liquid and different polymers were added (see Supplementary **Fig. 3**). The results show that the ILP composites with three kinds of polymers present different distributions and morphologies on the MF sponge, which may have influence on filtration performance of the filters. Besides, the polymer species would affect physicochemical properties of the ILP composites such as hydrophilicity, electroconductivity, and plasticity etc., which may be also the factors influencing the purification ability of the filters. The corresponding supplementary description has been added in the revised manuscript.

6. The composition in the smoke of a burned cigarette is complex. Authors should provide evidence to support their claiming of “particles in the size range from <50 nm to $>10\ \mu\text{m}$ ”, since filtrated nanoparticles are an essential for their topic of the manuscript.

Answer: According to your suggestion, we collected the particles produced by the smoke of burned cigarette and determined the size distribution by DLS and laser particle analysis (see Supplementary **Figs. 16** and **17**). The results show that the particles from the smoke possess a size range from a dozen nanometers to tens of microns. The corresponding statement was added in the “Method” section of the revised manuscript.

7. The lifetime is a very important factor for an air filter. The authors only carried out a "long-term" test for 15 hours which is not sufficient and convincing. Even in such a 15 h long-term test, the filtration efficiency decreased more than 1 %. If the filter is test for longer time, very likely the filtration efficiency would decrease more. The authors are suggested to conduct a longer-term test of at least a month to illustrate the stability of their filters.

Answer: We agree with the referee that the stability is of great significance for practical application of materials. Therefore, we further developed a continuous filtration test to explore long-term stability of the mask manufactured by ILP@MF filters. As the figure we added shown in **Fig. 5d**, the ILP@MF filter presents a good

stability during the purification test for at least three weeks. Unfortunately, after a long-time uninterrupted running, there are severe mechanical breakdown on our air compressor. The researchers who were 24-hour on duty also caught cold. Therefore, we feel sincerely sorry that we are not able to continue the test to a month. We think that the test at this stage may have certain reference value for the stability of the filters. We hope the results of these data will answer the reviewer's question.

Point-by-Point Response to Reviewers
REVIEWERS' COMMENTS:

Reviewer #1 (Remarks to the Author):

Responses to Reviewer #1

The authors have addressed all of the prior comments. I am especially appreciative of the effort to have professional editing help to improve the readability. I have two follow-on recommendations:

1. To point the reader to the Methods section near the end of the manuscript, I recommend that the following sentence be added to line 102 (or elsewhere): “More details of the methods employed are included in the Methods section near the end of the paper.”

Answer: We feel great thanks for the referee’s further recommendations. The sentence was added to line 102.

2. The table comparing costs at the end of the authors’ response is very useful and should be added to the Discussion section of the paper. In addition, a short summary table comparing the removal efficiencies of the different mask types should be added to the Discussion section.

Answer: According to your suggestion, a table containing the comparison of costs and removal efficiencies of the different mask types was added to the Discussion section, and the corresponding statement was also added.

Responses to Reviewer #2

The authors addressed the prior concerns. The additional work they did to improve the manuscript is notable. Here are a couple of further recommendations.

1. The authors added a 21-day stability test (Figure 5d). Rather than just stating that the removal efficiencies stayed above a certain value, it would be helpful to report in the text the values after 0, 7, 14, and 21 days (including the +/- values), since it is hard to read them from the plot. Also, if not already done, the +/- values should be defined. For example, are they standard derivations, 90% confidence intervals, or something else?

Answer: We appreciate the referee for the further recommendations. According to your suggestion, the values of removal efficiencies after 0, 7, 14, and 21 days were added to the text to help reader know the variation during the test. Furthermore, the +/- values represents the standard deviation of three replicate measurements, which are defined in the revised manuscript.

2. The authors measured the particle size distribution of cigarette smoke, and added it to the Supplementary Material (Figures 16 and 17). I would expect there to be literature data already available for such an important subject, so adding a comparison and reference is recommended.

Answer: We agree with the referee that a literature data is helpful. A comparison with available literature data was added to the text, and corresponding references (*ACS Nano* **11, 6211–6217 (2017), *Nat. Commun.* **6**, 6205 (2015), and *Nano Lett.* **17**, 4339–4346 (2017)) were recommended.**